# On the SAC-BL Algorithm for Anomaly Detection

**Xinsong Ma**
School of Computer Science
Wuhan University
maxinsong1018@gmail.com

**Jie Wu**
School of Computer Science
Wuhan University
biandekeren@gmail.com

**Weiwei Liu** *
School of Computer Science
Wuhan University
liuweiwei863@gmail.com

## Abstract

Visual anomaly detection is significant in safety-critical and reliability-sensitive scenarios. Prior studies mainly emphasize the design and training of scoring functions, while little effort has been devoted to constructing decision rules based on these score functions. A recent work Ma et al. (2025b) highlights this issue and proposes the SAC-BL algorithm to address it. This method consists of a strong anomaly constraint (SAC) network and a betting-like (BL) algorithm serving as the decision rule. The SAC-BL algorithm can control the false discovery rate (FDR). However the performance of SAC-BL algorithm on anomalous examples, or its false positive rate (FPR), has not been thoroughly investigated. This paper provides a deeper analysis of this problem and explores how to theoretically reduce its FPR. First, we show that as the number of testing examples tends to infinity, the SAC-BL algorithm performs well on abnormal data if the scores follow the generalized Gaussian-like distribution family. But such conditions about the number of testing examples and the distribution of scores are overly restrictive for the real-world applications. So, we attempt to decrease the FPR of the SAC-BL algorithm under the condition of finite samples for practical anomaly detection. To this end, we redesign the BL algorithm by incorporating a randomization strategy and propose a novel stochastic BL (SBL) algorithm. The combination of the SAC network and the SBL algorithm yields our method, SAC-SBL. Theoretical results show that the SAC-SBL algorithm can achieve smaller FPR than SAC-BL algorithm while controlling its FDR. Finally, extensive experimental results demonstrate the superiority of our method over SAC-BL algorithm on multiple visual anomaly detection benchmarks.

## 1 Introduction

Visual anomaly detection (AD) (Haselmann et al., 2018) is a critical task in computer vision, where the objective is to identify and spatially localize regions in visual data that deviate from normal patterns (Gong et al., 2019; Tien et al., 2023). The AD has wide-ranging applications, including industrial quality control (Bergmann et al., 2019a), medical image analysis (Huang et al., 2024), video surveillance (Lv et al., 2023) and autonomous driving (Huang et al., 2020). Unlike standard supervised learning problems, The AD task is characterized by a scarcity or complete absence of labeled anomalous data, due to the inherent rarity, unpredictability, and diversity of anomalies in real-world scenarios (Bergmann et al., 2019b). Consequently, most existing approaches adopt an

---

*Corresponding author

39th Conference on Neural Information Processing Systems (NeurIPS 2025).

unsupervised learning paradigm, where models are trained solely on normal samples and then identify the anomalies during inference based on the learned distribution information of normality (Wang et al., 2021; Bergmann et al., 2019c; Deng, Li, 2022).

A substantial body of methods have been proposed to tackle the challenge of the AD. These methods can be grouped into four principal categories: knowledge distillation-based methods (Deng, Li, 2022; Tien et al., 2023), synthesizing-based methods (Li et al., 2021; Yan et al., 2021), embedding-based methods (Defard et al., 2020; Roth et al., 2022) and reconstruction-based method (Akcay et al., 2018; Bergmann et al., 2019c). The existing approaches mainly focus on designing the powerful score functions to learn the discriminative information from normal data. However, these methods neglect the deep investigation for the decision rules based on the proposed score functions. A recent work Ma et al. (2025b) points out this issue and tackle it from statistical perspective. Concretely, Ma et al. (2025b) frames the AD task as a hypothesis testing problem and proposes a novel SAC-BL algorithm to address it. The SAC-BL algorithm consists of a strong anomaly constraint (SAC) network and a betting-like (BL) algorithm serving as the decision rule. Theoretically, the SAC-BL algorithm can control false discovery rate (FDR) at prescribed level. However, the performance of the SAC-BL algorithm on anomalous examples, or its false positive rate (FPR), has not been thoroughly analyzed. Factually, only controlling the FDR might result in a poor performance on anomalous data in some worst-case scenario. For example, we consider a trivial decision rule $\phi(\cdot)$ which directly accepts all null hypotheses, equivalent to classify all testing examples as normal data. Obviously, the FDR of decision rule $\phi(\cdot)$ is zero, but its FPR is one. It is therefore imperative to conduct an in-depth study on the FPR of SAC-BL algorithm. Besides, a review (Pang et al., 2021) highlights that existing anomaly detection methods—particularly unsupervised methods—often suffer from high FPR, and how to reduce FPR remains one of the most important yet challenging problems in the field. Then, one natural question arises:

> *How to theoretically reduce the FPR of SAC-BL algorithm while controlling the FDR at prescribed level?*

This paper attempts to address the problems mentioned above.

Based on the analytical framework in Ingster, Suslina (2003); Donoho, Jin (2004); Jin, Ke (2016), we find that as the number of testing examples tends to infinity, the SAC-BL algorithm performs well on anomalous data, provided that the score distribution belongs to the generalized Gaussian-like family. However, such assumptions are often overly restrictive for real-world applications. On the one hand, real-world data distributions are complex and typically unknown, making it difficult to satisfy specific distributional requirements. On the other hand, the number of available testing samples is often limited. For example, in the widely used AD benchmark MVTec (Bergmann et al., 2019b), the testing set for the class "Pill" contains only 167 images. Hence, it is necessary to enhance the performance of SAC-BL algorithm on the anomalous data under the condition of finite examples for the real-world AD task. To this end, we redesign the BL algorithm by incorporating a randomization strategy and propose a novel stochastic BL (SBL) algorithm. The combination of the SAC network and the SBL algorithm yields our method, SAC-SBL. Theoretical results show that SAC-SBL algorithm can achieve smaller FPR than the SAC-BL algorithm while controlling its FDR at the prespecified level. Finally, we conduct extensive experiments to verify the effectiveness of SAC-SBL algorithm. For example, compared with the SAC-BL algorithm, our method reduces the image-level FPR from $43.26\%$ to $29.08\%$ while achieving the same TPR for the class "Pill" in MVTec.

We summarize our main contributions as follows.

1. We demonstrate that as the number of testing examples tends to infinity, the SAC-BL algorithm achieves a well performance on anomalous testing examples if the distribution of scores belongs to the generalized Gaussian-like distribution family.

2. To improve the performance of SAC-BL algorithm on anomalous examples in practical AD task, we propose a novel SAC-SBL algorithm which is based on a randomization strategy for the p-values. Theoretically, our proposed method can reduce the FPR of SAC-BL algorithm while controlling its FDR at prescribed level.

3. Extensive experimental results demonstrate the superiority of our method over SAC-BL algorithm on multiple visual anomaly detection benchmarks.

## 2 Background

Different from previous literature, Ma et al. (2025b) studies the AD problem from the perspective of multiple hypothesis testing, and propose the SAC-BL algorithm to tackle it. We first introduce the hypothesis testing framework introduced by Ma et al. (2025b) for the AD task. To avoid confusion, we use the same mathematical notations as Ma et al. (2025b) in our paper. Denote by $\mathcal{X}$ the feature space of the normal examples, and $\mathcal{X}$ follows the underlying distribution $\mathcal{D}$. In most cases, $\mathcal{D}$ is unknown. Given a testing set $\mathcal{T}^{test} = \{X_1^{test}, X_2^{test}, X_3^{test}, \cdots, X_n^{test}\}$ [2], Ma et al. (2025b) frames the AD task as the following multiple hypothesis testing problem:

$$
\begin{aligned}
H_{1;0} &: X_1^{test} \sim \mathcal{D}, \quad H_{1;1} : X_1^{test} \not\sim \mathcal{D} \\
H_{2;0} &: X_2^{test} \sim \mathcal{D}, \quad H_{2;1} : X_2^{test} \not\sim \mathcal{D} \\
&\qquad\qquad \vdots \\
H_{n;0} &: X_n^{test} \sim \mathcal{D} \quad H_{n;1} : X_n^{test} \not\sim \mathcal{D}
\end{aligned}
\tag{1}
$$

where $H_{i;0}$ and $H_{i;1}$ are called null hypothesis and alternative/non-null hypothesis, respectively. In the context of anomaly detection, if $H_{i,0}$ is rejected, we declare that $X_i^{test}$ is anomalous.

In statistics, the decision to accept or reject the null hypothesis is determined by the concept of *p-value*. Its general definition is presented as follows.

**Definition 2.1.** (**P-value (Casella, Berger, 2002)**) Given a sample $\widetilde{X}$ [3]. A statistic $p(\widetilde{X})$ is called p-value corresponding to the null hypothesis $H_0$, if $p(\widetilde{X})$ satisfies

$$
\mathbb{P}[p(\widetilde{X}) \leq t | H_0] \leq t
\tag{2}
$$

for every $0 \leq t \leq 1$.

A small p-value usually provides strong evidence against the null hypothesis. If the cumulative distribution function $F(\cdot)$ of testing statistic $T$ for null hypothesis is known, then the corresponding p-value can be defined as

$$
p(\hat{T}) = \mathbb{P}(T > \hat{T}) = 1 - F(\hat{T}),
$$

where $\hat{T}$ is the observation of $T$. It is easy to demonstrate that $p(\hat{T})$ satisfies the condition in Eq. (2). It is noteworthy that the p-value has clear statistical interpretation. For example, suppose the p-value of a anomalous testing example $X_i^{test}$ is 0.01. This means that for any coming testing example $X_j^{test}$, the probability that $X_j^{test}$ is more anomalous than $X_i^{test}$ is 0.01. In other words, it is extremely difficult to find a more anomalous example than $X_i^{test}$. Hence, we are highly confident that $X_i^{test}$ is abnormal.

It is known that the probability of type-I error should be controlled at the prescribed significant level in single hypothesis testing. Similarly, in multiple hypothesis testing, the false discovery rate (FDR), as the generalization of probability of type-I error, should be controlled. The statistical advantages of FDR have been detailedly discussed in (Benjamini, Hochberg, 1995; Benjamini, Yekutieli, 2001).

Given the null hypotheses $\{H_{1;0}, H_{2;0}, \cdots, H_{n;0}\}$, let $\mathcal{R}$ be the set of indices of the rejected null hypotheses. Similarly, denote by $\mathcal{H}_0$ and $\mathcal{H}_1$ the set of indices for the true null hypotheses and false null hypotheses for $\{H_{1;0}, H_{2;0}, \cdots, H_{n;0}\}$, respectively. Besides, let $n_0 = |\mathcal{H}_0|$ be the number of true null hypotheses. In statistics, if one null hypothesis is rejected, it is said to make a discovery. FDR is the expected proportion of false discoveries among the rejected hypotheses.

**Definition 2.2** (**FDR(Benjamini, Hochberg, 1995)**). Dnote by $V$ the number of true null hypotheses rejected for the hypotheses $H_{1;0}, H_{2;0}, \cdots, H_{n;0}$. Additionally, let $U$ be the number of rejected hypotheses. The false discovery proportion (FDP) is defined as:

$$
\text{FDP} = \begin{cases} V/U, & \text{if } U > 0, \\ 0, & \text{otherwise.} \end{cases}
$$

The expectation of FDP is defined as the FDR, namely

$$
\text{FDR} = \mathbb{E}(\text{FDP}) = \mathbb{E}\left[\frac{|\mathcal{R} \cap \mathcal{H}_0|}{\max\{1, |\mathcal{R}|\}}\right].
$$

---

[2] $X_i^{test}$ is a image or pixel.

[3] A sample means a sequence of examples.

We given a new insight for controlling the FDR. Following Ma et al. (2025b), the normal data is set to be positive. The FDP is closely related to the TPR and FPR. Using the notations of confusion matrixm, the FDP is expressed as

$$\text{FDP} = \frac{FN}{FN + TN} = \frac{1}{1 + \frac{TN}{FN}}$$
$$= \frac{1}{1 + \frac{N-FP}{P-TP}} = \frac{1}{1 + \frac{P}{N} \cdot \frac{1-FPR}{1-TPR}}. \tag{3}$$

Note that $P = |\mathcal{H}_0|$ and $N = |\mathcal{H}_1|$. For a given testing set, $P$ and $N$ are fixed. It is well-known that there is a tradeoff between the detection performance of normal and abnormal examples for a trained score functions. Therefore, we cannot only consider the true positive rate (TPR) or false positive rate (TPR) when designing the AD algorithm. Factually, an ideal AD algorithm should achieve low FPR while maintaining a high TPR, which leads to a small FDP based on Eq. (3). Thus, controlling FDR tends to achieve a well tradeoff between the detection performance of both normal and abnormal examples if $\mathcal{R} \neq \emptyset$.

To control the FDR for the AD task, Ma et al. (2025b) proposes a new SAC-BL algorithm, which consists of a SAC network serving as the score function and a BL algorithm as the decision rule. The SAC network is composed of two parts: a reconstrunction network (R-net) and a discriminative network (D-net). The core of SAC network is to apply strong constraints for the training process of discriminator by hard pairs generated from reconstruction network, which can improve the discriminative capability for weak anomalies. The details about SAC network can be found in Section 4 of Ma et al. (2025b). The BL algorithm is defined as follows.

**Definition 2.3** (**BL algorithm (Ma et al., 2025b)**). Given the p-values $p_1, p_2, \cdots, p_n$ corresponding to the null hypotheses $H_{1,0}, H_{2,0}, \cdots, H_{n,0}$, let $p_{(i)}$ be the $i$-th order statistics from the smallest to the largest. For a pre-specified level $\alpha \in (0, 1)$, define

$$i_{BL}^* := \max \left\{ i \in [1:n] : \frac{1}{p_{(i)}^\gamma} \geq \frac{n}{\delta\alpha i} \right\} = \max \left\{ i \in [1:n] : p_{(i)}^\gamma \leq \frac{\delta\alpha i}{n} \right\} \tag{4}$$

where $\gamma, \delta$ are two positive real numbers and satisfy $\gamma + \delta \leq 1$. Then, the null hypothesis $H_{(i),0}$ is rejected if $i \leq i_{BL}^*$.

In statistics, $\alpha$ is usually set to 0.05. Eq. (4) indicates that the BL algorithm rejects the null hypothesis $H_{i,0}$ if $p_i^\gamma \leq \frac{\alpha\delta}{n} i_{BL}^*$ and $|\mathcal{R}| = i_{BL}^*$. In practice, for a testing example $X_i^{test}$, if the distribution of testing statistic (or scores) is known, the p-value $p_i$ corresponding to null hypothesis $H_{i;0}$ can be presented as $p_i = 1 - F(s(X_i))$ where $F(\cdot)$ is the cumulative distribution function of the scores and $s(\cdot)$ is the score function. If the distribution information is unknown, the computation of the p-value relies on a calibrated set consisting of normal examples. Specifically, given a calibrated set $\mathcal{T}^{cal} = \{X_1^{cal}, X_2^{cal}, \ldots, X_k^{cal}\}$, $p_i$ is presented as

$$p_i = p(X_i^{test}) = \frac{\sum_{j=1}^k \mathbf{1}(s(X_j^{cal}) \geq s(X_i^{test})) + 1}{k + 1}. \tag{5}$$

## 3 Asymptotic FPR of SAC-BL algorithm

False positive rate (FPR) is a significant evaluation criterion for AD task, which can be expressed as

$$\text{FPR} = \frac{FP}{FP + TN} = \frac{|\mathcal{R}^c \cap \mathcal{H}_1|}{|\mathcal{H}_1|}.$$

Although Ma et al. (2025b) has proved that the SAC-BL algorithm can control FDR. However, relatively little is known about the theoretical properties of FPR of SAC-BL algorithm. In this section, we aim to analyze the asymptotic performance of its FPR. Our analytical framework follows Donoho, Jin (2004); Jin, Ke (2016)

In the vast majority of theoretical literature on multiple hypothesis testing (Benjamini, Hochberg, 1995; Benjamini, Yekutieli, 2001; Storey, 2002; Blanchard, Roquain, 2008), the p-values are assumed to be available, equivalently, the distribution of test statistic for each null hypothesis is known. In this

situation, the p-value can be represented as $p_i = \Psi(\tilde{T}_i)$ [4] where $\tilde{T}_i$ is the observation of test statistic $T_i$ corresponding to hypothesis $H_i$ and $\Psi(\cdot)$ is the survival function of test statistic. For analytical simplicity, we suppose that the observation $\tilde{T}_1, \tilde{T}_2, \cdots, \tilde{T}_n$ are continuous random variables. We first define a function class as follows.

$$\mathcal{F} = \left\{ \Psi(t) : \lim_{t \to \infty} \frac{-\log \Psi(t)}{t^\tau} = \frac{1}{\tau}, \ \tau > 1 \right\}.$$

Based on the function class $\mathcal{F}$, we have the following definition.

**Definition 3.1** (Generalized Gaussian-like Distribution Family)**.** A random variable $X$ is said to follow the generalized Gaussian-like distribution with the location $\mu$ and the degree $\tau$, denote $X \sim G(\mu, \tau)$, if the survival function $\Psi(\cdot)$ of $X$ satisfies $\Psi(t - \mu) \in \mathcal{F}$.

It is easy to verify that Gaussian distribution belongs to this distribution family. Similar to Donoho, Jin (2004); Ingster, Suslina (2003); Jin, Ke (2016), our analytical framework relies on the above generalized Gaussian-like distribution family. Specifically, we assume that the test statistic $T_i$ [5] corresponding to the hypothesis $H_i$ satisfies:

$$T_i \sim \begin{cases} G(0, \tau), & \text{if } i \in \mathcal{H}_0 \\ G(\mu, \tau), & \text{if } i \in \mathcal{H}_1 \end{cases} \tag{6}$$

Following Donoho, Jin (2004), we set $\mu = (\tau r \log n)^{1/\tau}$. Besides, we focus on the sparse region in which the number of true null hypothesis is larger than that of the true alternative hypothesis (Donoho, Jin, 2006). In this case, $n_1 = |\mathcal{H}_1| = n^{1-\beta}$ where $\beta < r < 1$ (note that $\frac{n_1}{n} \to 0$). Based on the framework above, we derive the asymptotic FPR of SAC-BL algorithm.

**Theorem 3.2.** *Suppose that the test statistic $T_i$ corresponding to $H_i$ satisfies the condition in Eq. (6) for $i \in [n]$. Then, for the p-values $p_1 = \Psi(\tilde{T}_1), p_2 = \Psi(\tilde{T}_3), \cdots, p_n = \Psi(\tilde{T}_n)$, the FPR of SAC-BL algorithm converges to zero in probability, namely*

$$\text{FPR}_{BL} = \frac{FP}{FP + TN} \to 0 \quad \text{in probability}$$

*as $n \to \infty$.*

The proof of Theorem 3.2 is presented in Appendix B.1. Based on the Theorem 3.2, we have the following theoretical result.

**Theorem 3.3.** *Suppose that the conditions in Theorem 3.2 hold. Then, we have*

$$\frac{FP}{FP + TN} \xrightarrow{P} 0 \quad \text{if and only if} \quad \lim_{n \to \infty} \mathbb{E}\left( \frac{FP}{FP + TN} \right) \to 0.$$

The proof of Theorem 3.3 is presented in Appendix B.2.

Theorem 3.2 indicates that as the number of testing samples tends to infinity, the SAC-BL algorithm performs well on anomalous examples, provided that the distribution of the scores belongs to the generalized Gaussian-like distribution family. However, such conditions are overly restrictive in real-world applications, as the distribution of real-world data is often complex and typically unknown. Besides, for the widely used AD benchmark MVTech (Bergmann et al., 2019b), the testing set for the class "Pill" only contains 167 images and the SAC-BL attains the image-level FPR of merely $43\%$ on it (see Table 1). Hence, we need to explore how to improve the performance of SAC-BL algorithm on the anomalous data under the condition of finite examples for the practical AD task.

## 4   Decreasing the FPR of SAC-BL algorithm

In this section, we explore how to reduce the FPR of SAC-BL algorithm while controlling its FDR at the prescribed level. Our main focus is to redesign the BL algorithm.

---

[4]Under the null hypothesis, all of test statistic have the same function. So, we use the same survival function.

[5]Note that the observations of test statistic can be regarded as its sample, thus the test statistic and its observations have the same distribution.

Note that $FPR = \frac{FP}{FP+TN} = \frac{|\mathcal{R}^c \cup \mathcal{H}_1|}{|\mathcal{H}_1|}$. where $\mathcal{R}^c$ is the complement of $\mathcal{R}$. For a given testing set, $FP + TN$ is equal to the number of true anomalous examples in testing set. Therefore, we can reduce the FPR by reducing FP, equivalent to increasing the number of rejecting null hypotheses $|\mathcal{R}|$. Recall the BL algorithm: $i^*_{BL} := \max\left\{ i \in [1:n] : p^\gamma_{(i)} \leq \frac{\delta\alpha i}{n} \right\}$. The null hypothesis $H_{i,0}$ is rejected if and only if $p^\gamma_{(i)} \leq \frac{\delta\alpha}{n} i^*_{BL}$. Denote

$$\xi_{BL} = \frac{\delta\alpha}{n} i^*_{BL}, \qquad \xi_i = \frac{\delta\alpha i}{n}, \qquad \mathcal{C} = \left\{ \frac{1\delta\alpha}{n}, \frac{2\delta\alpha}{n}, \cdots, \frac{n\delta\alpha}{n} \right\}.$$

Note that $\xi_{BL}$ only take the value in $\mathcal{C}$. For a given p-value $p_i$ satisfying $p^\gamma_i > \frac{\delta\alpha}{n}\xi_{BL}$, our idea is to develop a strategy that changes the value of $p_i$ to $\tilde{p}_i$ such that $\tilde{p}^\gamma_i = \xi_{BL}$ without violating the core conditions of controlling FDR for the SAC-BL algorithm. Then we could increase the number of rejecting null hypotheses and further reduce the FPR of SAC-BL algorithm.

Denote by $\Lambda$ the random variable uniformly distributed on $(0,1)$ independent of p-values and we define
$$\Upsilon_i(x) = x^\gamma \cdot \mathbf{1}\left(x^\gamma \leq \xi_i\right) + \xi_i \cdot \mathbf{1}\left(\Lambda\xi_i < \Lambda x^\gamma \leq \xi_i\right).$$
Moreover, for each $i \in [n] := \{1,2,3,\cdots,n\}$, we denote

$$\epsilon_i = \sum_{j=1}^n \mathbf{1}\left(p_j \leq p_i\right), \qquad \kappa(i) = \arg\max_{j \geq \epsilon_i} \frac{j}{p^\gamma_j}.$$

If there are multiple indices that satisfy the argmax, we take the largest index. Then, we define the stochastic BL (SBL) algorithm as follows.

**Definition 4.1.** (**Stochastic BL Algorithm**) Given the random variable $\Lambda$ uniformly distributed on $(0,1)$, p-values $p_1, p_2, \cdots, p_n$ corresponding to the null hypotheses $H_{1,0}, H_{2,0}, \cdots, H_{n,0}$ and the prespecified level $\alpha \in (0,\ 1)$. For each $i \in [n]$, define

$$\Upsilon_{\kappa(i)}(p_i) = p^\gamma_i \cdot \mathbf{1}\left(p^\gamma_i \leq \xi_{\kappa(i)}\right) + \xi_{\kappa(i)} \cdot \mathbf{1}\left(\Lambda\xi_{\kappa(i)} < \Lambda p^\gamma_i \leq \xi_{\kappa(i)}\right)$$

and

$$i^*_{BL} := \max\left\{ i \in [1:n] : \frac{1}{p^\gamma_{(i)}} \geq \frac{n}{\delta\alpha i} \right\} = \max\left\{ i \in [1:n] : p^\gamma_{(i)} \leq \frac{\delta\alpha i}{n} \right\},$$

where $\gamma, \delta$ are two positive real numbers and satisfy $\gamma + \delta \leq 1$. Then, the null hypothesis $H_{i;0}$ is rejected if

$$\Upsilon_{\kappa(i)}(p_i) \leq \frac{\delta\alpha}{n} i^*_{BL}.$$

The SBL algorithm in Definition 4.1 is mainly used to derive our theoretical results. Additionally, we have another simple version of SBL algorithm.

**Definition 4.2.** (**Stochastic BL algorithm: version 2**) Given the p-values $p_1, p_2, \cdots, p_n$ corresponding to the null hypotheses $H_{1,0}, H_{2,0}, \cdots, H_{n,0}$, let $p_{(i)}$ be the $i$-th order statistics from the smallest to the largest. Denote by $\Lambda$ the random variable uniformly distributed on $(0,1)$. For a pre-specified level $\alpha \in (0,\ 1)$, define

$$i^*_{SBL} := \max\left\{ i \in [1:n] : \frac{1}{p^\gamma_{(i)}} \geq \frac{n\Lambda}{\delta\alpha i} \right\} \tag{7}$$

where $\gamma, \delta$ are two positive real number and satisfies $\gamma + \delta \leq 1$. Then, the null hypothesis $H_{(i),0}$ is rejected if $i \leq i^*_{SBL}$.

The Following theory reveals the relation between the SBL algorithms in Definition 4.1 and the one in Definition 4.2.

**Theorem 4.3.** *The SBL algorithm in Definition 4.1 is equivalent to the one in Definition 4.2.*

The proof of Theorem 4.3 is presented in Appendix B.3. Obviously, the SBL algorithm in Definition 4.2 has less computation steps than the one in Definition 4.1, and thus is more suitable for the practical

---
**Algorithm 1:** Practical SAC-SBL Algorithm
---
**Input:** Training set $\mathcal{T}^{tra}$, calibrated set $\mathcal{T}^{cal} = \{X_1^{cal}, X_2^{cal}, \ldots, X_k^{cal}\}$ testing set
$\qquad \mathcal{T}^{test} = \{X_1^{test}, X_2^{test}, \ldots, X_n^{test}\}$, prescribed level $\alpha \in (0, 1)$, generator of synthetic
$\qquad$ anomalous example $G(\cdot)$.

1 Utilize $G(\cdot)$ to construct the new training set $\mathcal{T}^{mix}$ based on the training set $\mathcal{T}^{tra}$.
2 Train the SAC network on $\mathcal{T}^{mix}$, and then obtain the score function $s(\cdot)$ for images or pixels.
3 Calculate the empirical p-value corresponding to $X_i^{test}$:

$$p_i = p(X_i^{test}) = \frac{\sum_{j=1}^{k} \mathbb{1}(s(X_j^{cal}) \geq s(X_i^{test})) + 1}{k + 1}. \tag{8}$$

4 Draw a sample $\Lambda$ from the uniform distribution on $(0, 1)$.
5 Determine the index $i_{BL}^*$:

$$i_{SBL}^* := \max\left\{ i \in [n] : \frac{1}{p_{(i)}^\gamma} \geq \frac{n\Lambda}{\delta\alpha i} \right\}$$

**Output:** Declare that $X_{(i)}^{test}$ is anomalous if $i \leq i_{SBL}^*$.

---

applications. The version of SBL algorithm in Definition 4.1 is used to establish the connection between the SBL algorithm and BL algorithm for the theoretical analysis.

In the context of anomaly detection, the underlying distribution information of normal data is usually unknown. Therefore, we can use the method in Eq. (5) to compute the p-values in practice. Combining the SAC network proposed by Ma et al. (2025b) and the SBL algorithm in Definition 4.2 yields our method, SAC-SBL. Its detailed steps are presented in Algorithm 1.

Now we present our core theoretical results to demonstrate the superiority of SAC-SBL algorithm over the SAC-BL algorithm in terms of FPR while controlling the FDR at the prescribed level.

**Theorem 4.4.** *Given the random variable $\Lambda$ uniformly distributed on $(0, 1)$, the p-values $p_1, p_2, \cdots, p_n$ corresponding to the null hypotheses $H_{1,0}, H_{2,0}, \cdots, H_{n,0}$ in Eq. (8) and the pre-specified level $\alpha \in (0, 1)$. The following conclusions hold:*

*1. the FDR of SAC-SBL algorithm satisfies*

$$FDR_{SAC-SBL} \leq \alpha;$$

*2. For the index sets $\mathcal{R}_{SAC-BL}$ and $\mathcal{R}_{SAC-SBL}$, we have*

$$\mathbb{P}\left(\mathcal{R}_{SAC-BL} \subseteq \mathcal{R}_{SAC-SBL}\right) = 1;$$

*3. If there exists a p-value $p_i$ which satisfies $\mathcal{P}\left(p_i^\gamma < \frac{\delta\alpha i_{BL}^*}{n}\right) > 0$, then we have*

$$\mathbb{P}\left(|\mathcal{R}_{SAC-BL}| < |\mathcal{R}_{SAC-SBL}|\right) > 0.$$

The proof of Theorem 4.4 is presented in Appendix B.4. Theorem 4.4 suggests that the SAC-SBL algorithm does not reject fewer null hypotheses than the SAC-BL algorithm almost surely while controlling the FDR at the prescribed level. In other words, SAC-BL algorithm tends to classify more testing examples as anomalous data. Based on Theorem 4.4, we can easily obtain the following theoretical results.

**Theorem 4.5.** *Suppose the conditions in Theorem 4.4 hold, then the FPR of the SAC-BL algorithm and that of SAC-SBL algorithm satisfy*

$$\mathbb{P}\left(FPR_{SAC-BL} \geq FPR_{SAC-SBL}\right) = 1.$$

*Moreover, if there exists a p-value $p_i$ which satisfies $\mathcal{P}\left(p_i^\gamma < \frac{\delta\alpha i_{BL}^*}{n}\right) > 0$, then we have*

$$\mathbb{P}\left(FPR_{SAC-BL} > FPR_{SAC-SBL}\right) > 0.$$

Table 1: Experimental results (%) on **MVTec**. We compare the performance between SAC-BL algorithm and SAC-SBL based on the same trained SAC network. ↑ indicates larger values are better and vice versa.

| Category | Method | Image-level | | | Pixel-level | | |
|---|---|---|---|---|---|---|---|
| | | TPR ↑ | FPR ↓ | F1-score ↑ | TPR ↑ | FPR ↓ | F1-score ↑ |
| Capsule | SAC-BL | 100.0 | 35.78 | 54.12 | 99.25 | 41.82 | 93.24 |
| | SAC-SBL | 100.0 | 26.61 | 61.33 | 99.25 | 34.77 | 99.43 |
| Bottle | SAC-BL | 95.00 | 3.17 | 92.68 | 95.40 | 3.63 | 97.42 |
| | SAC-SBL | 95.00 | 0.00 | 97.44 | 95.40 | 3.62 | 97.53 |
| Carpet | SAC-BL | 100.0 | 15.73 | 80.00 | 90.39 | 4.94 | 94.87 |
| | SAC-SBL | 92.14 | 4.49 | 83.64 | 90.37 | 4.93 | 94.90 |
| Leather | SAC-BL | 100.0 | 4.35 | 94.12 | 98.19 | 4.24 | 99.06 |
| | SAC-SBL | 100.0 | 0.00 | 100.0 | 98.21 | 4.16 | 99.08 |
| Pill | SAC-BL | 100.0 | 43.26 | 46.02 | 95.28 | 4.78 | 97.42 |
| | SAC-SBL | 100.0 | 29.08 | 55.91 | 95.27 | 4.77 | 97.49 |
| Transistor | SAC-BL | 100.0 | 30.00 | 90.91 | 96.63 | 42.62 | 90.18 |
| | SAC-SBL | 100.0 | 30.00 | 90.91 | 96.62 | 31.58 | 97.22 |
| Tile | SAC-BL | 100.0 | 1.19 | 98.51 | 95.01 | 0.77 | 97.38 |
| | SAC-SBL | 100.0 | 0.00 | 100.0 | 95.01 | 0.77 | 97.38 |
| Cable | SAC-BL | 98.28 | 21.74 | 84.44 | 93.33 | 19.89 | 95.96 |
| | SAC-SBL | 96.55 | 15.22 | 87.50 | 92.85 | 16.95 | 96.01 |
| Zipper | SAC-BL | 100.0 | 0.84 | 98.46 | 97.32 | 3.83 | 98.60 |
| | SAC-SBL | 100.0 | 0.84 | 98.46 | 97.32 | 3.83 | 98.60 |
| Toothbrush | SAC-BL | 100.0 | 3.33 | 96.00 | 94.04 | 1.76 | 96.90 |
| | SAC-SBL | 100.0 | 0.00 | 100.00 | 94.05 | 1.71 | 96.92 |
| Metal_nut | SAC-BL | 100.0 | 1.08 | 97.78 | 96.31 | 3.63 | 97.64 |
| | SAC-SBL | 100.0 | 0.00 | 100.0 | 96.07 | 3.28 | 97.78 |
| Hazelnut | SAC-BL | 100.0 | 1.43 | 98.77 | 98.99 | 10.24 | 95.27 |
| | SAC-SBL | 100.0 | 0.00 | 100.0 | 98.92 | 8.63 | 99.35 |
| Screw | SAC-BL | 100.0 | 72.27 | 48.81 | 99.40 | 12.80 | 95.67 |
| | SAC-SBL | 100.0 | 68.91 | 50.00 | 99.40 | 8.77 | 99.69 |
| Grid | SAC-BL | 100.0 | 19.30 | 79.25 | 97.56 | 0.63 | 98.76 |
| | SAC-SBL | 100.0 | 1.75 | 97.67 | 97.57 | 0.61 | 98.77 |
| Wood | SAC-BL | 100.0 | 8.33 | 88.37 | 92.69 | 7.09 | 94.92 |
| | SAC-SBL | 100.0 | 0.00 | 100.0 | 92.70 | 6.99 | 96.07 |
| Average | SAC-BL | **99.55** | 17.45 | 83.22 | **95.99** | 10.85 | 96.22 |
| | SAC-SBL | 98.91 | **11.79** | **88.19** | 95.93 | **9.03** | **97.75** |

The proof of theorem 4.5 is presented in Appendix B.5. Theorem 4.5 indicates that the FPR of the SAC-BL algorithm is not smaller than that of SAC-SBL algorithm almost surely. Besides, the condition "there exists a p-value $p_i$ which satisfies $\mathcal{P}\left(p_i^\gamma < \frac{\delta\alpha i_{BL}^*}{n}\right) > 0$" is nearly always satisfied in practice, since it only excludes the trivial situation where all hypotheses are rejected. Therefore, our proposed method can achieve a smaller FPR over the SAC-BL algorithm.

# 5 Experiment

In this section, we perform extensive comparison experiments to verify the effectiveness of our SAC-SBL method.

## 5.1 Experimental Settings

**Baseline.** We compare the decision-making performance of our proposed SAC-SBL method with the SAC-BL method (Ma et al., 2025b), using the default settings for SAC-BL.

**Datasets.** Experiments are conducted on three widely used anomaly detection datasets. The MVTec dataset (Bergmann et al., 2019b) consists of 5354 images across 15 object and texture categories. The VisA dataset (Zou et al., 2022) is an industrial anomaly dataset comprising 10821 images from 12 objects across 3 domains. The BTAD dataset (Mishra et al., 2021) includes 2830 images of 3 industrial products, showcasing body and surface defects.

**Evaluation Metrics.** We use the true positive rate (TPR), false positive rate (FPR) and f1-score (F1) to evaluate the effectiveness of the proposed method.

Table 2: Experimental results (%) on **VisA**. We compare the performance between SAC-BL algorithm and SAC-SBL based on the same trained SAC network. ↑ indicates larger values are better and vice versa.

| Category | Method | Image-level | | | Pixel-level | | |
|---|---|---|---|---|---|---|---|
| | | TPR ↑ | FPR ↓ | F1-score ↑ | TPR ↑ | FPR ↓ | F1-score ↑ |
| Candle | SAC-BL | 100.0 | 49.00 | 80.97 | 99.88 | 55.26 | 95.87 |
| | SAC-SBL | 100.0 | 46.00 | 83.30 | 99.88 | 52.26 | 99.90 |
| Capsules | SAC-BL | 100.0 | 52.00 | 70.59 | 98.39 | 9.83 | 99.16 |
| | SAC-SBL | 98.33 | 46.00 | 71.52 | 98.39 | 6.72 | 99.18 |
| Cashew | SAC-BL | 100.0 | 87.00 | 54.05 | 99.30 | 67.93 | 90.88 |
| | SAC-SBL | 100.0 | 85.00 | 54.05 | 99.25 | 64.51 | 99.24 |
| Chewinggum | SAC-BL | 100.0 | 15.00 | 86.96 | 94.38 | 4.94 | 97.10 |
| | SAC-SBL | 100.0 | 14.00 | 87.72 | 94.85 | 4.92 | 97.33 |
| Fryum | SAC-BL | 100.0 | 67.00 | 56.88 | 99.44 | 65.78 | 92.19 |
| | SAC-SBL | 100.0 | 57.00 | 63.69 | 99.42 | 65.39 | 98.94 |
| Macaroni1 | SAC-BL | 100.0 | 81.00 | 71.17 | 99.91 | 18.30 | 99.95 |
| | SAC-SBL | 100.0 | 79.00 | 76.68 | 99.91 | 18.21 | 99.95 |
| Macaroni2 | SAC-BL | 100.0 | 55.00 | 74.37 | 99.92 | 30.89 | 93.46 |
| | SAC-SBL | 100.0 | 48.00 | 80.65 | 99.92 | 25.98 | 99.95 |
| Pcb1 | SAC-BL | 98.00 | 69.00 | 70.41 | 95.98 | 13.36 | 97.92 |
| | SAC-SBL | 98.00 | 64.00 | 73.68 | 95.98 | 13.36 | 97.92 |
| Pcb2 | SAC-BL | 97.00 | 45.00 | 80.17 | 92.97 | 30.77 | 96.29 |
| | SAC-SBL | 96.00 | 31.00 | 83.56 | 92.98 | 30.69 | 96.33 |
| Pcb3 | SAC-BL | 99.01 | 56.00 | 76.82 | 99.41 | 39.24 | 99.60 |
| | SAC-SBL | 98.02 | 50.00 | 79.20 | 99.41 | 38.94 | 99.66 |
| Pcb4 | SAC-BL | 98.02 | 13.00 | 90.96 | 98.08 | 24.94 | 98.87 |
| | SAC-SBL | 97.03 | 7.00 | 95.15 | 98.08 | 24.92 | 98.95 |
| Pipe_fryum | SAC-BL | 100.0 | 100.0 | 50.00 | 99.34 | 73.15 | 91.75 |
| | SAC-SBL | 100.0 | 96.00 | 50.25 | 99.34 | 60.10 | 99.21 |
| Average | SAC-BL | **99.34** | 57.42 | 71.95 | 98.08 | 36.20 | 96.09 |
| | SAC-SBL | 98.95 | **51.92** | **74.95** | **98.12** | **33.83** | **98.88** |

**Implementation Details.** In SAC-SBL, we repeat the stochastic perturbation 100 times and take the meaning value. All experiments are conducted on a workstation with eight NVIDIA GeForce GTX 3090 GPUs and two 2.2GHZ Intel CPUs.

## 5.2 Experimental Results

We evaluate our proposed method against the SAC-BL algorithm on three datasets, demonstrating its superior performance and generalization. The image-level and pixel-level results on MVTec are presented in Table 1, and the results on VisA are shown in Table 2. because of the space limitation, the results in BTAD are reported in Appendix A. From the Tables 1 and 2, we can see: 1) Our mehtod achieves the comparable performance in terms of TPR. For example, in Table 1, our method have the same image-level TPR as the SAC-BL algorithm in 13 classe among 15 classes. On average, the image-level and pixel-level TPRs of two methods are also comparable. 2) Our method yields lower FPR and higher F1-scores across all classes, regardless of image-level or pixel-level evaluation. As shown in Table 1, compared to the SAC-BL algorithm, our method reduce the image-level FPR from $21.74\%$ to $15.22\%$, and improve the image-level F1-score from $84.44\%$ to $87.5\%$, the direct improvements of $6.52\%$ and $3.06\%$, respectively. 3) In those classes where the SAC-BL algorithm struggles to identify anomalous examples, our method can considerably reduce the FPR. For instance, For classes "Carpet" and "Pill", our method achieve the improvements of $11.24\%$ and $14.14\%$ in terms of image-level FPR. Overall, these experimental results demonstrate the superiority of our method over the SAC-BL algorithm.

## 6 Conclusion

In this paper, we focus on investigating the SAC-BL algorithm in terms of FPR. First, we demonstrate that the FPR of SAC-BL algorithm converges to zero in probability based on the generalized Gaussian-like distribution family. Then, we propose a novel SAC-SBL algorithm which can reduce the FPR of SAC-BL algorithm while controlling the FDR at the prescribed level. Finally, we conduct extensive experiments to verify the effectiveness of our proposed method.

## Acknowledgment

This work is supported by the National Natural Science Foundation of China under Grant 624B2106, the Key R&D Program of Hubei Province under Grant 2024BAB038, the National Key R&D Program of China under Grant 2023YFC3604702, the Fundamental Research Funds for the Central Universities under Grant 2042025kf0045.

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
