# OpenReview forum: "On the SAC-BL Algorithm for Anomaly Detection"
_NeurIPS.cc/2025/Conference — NeurIPS 2025 poster_

### Official Review · Reviewer_aLKU · 2025-06-23

**Clarity:** 3
**Significance:** 2
**Originality:** 3
**Rating:** 4
**Confidence:** 5

**Summary:**

In this paper, the authors provide a deeper analysis of this problem and explores how to theoretically reduce its False Positive rate. First, the authors show that as the number of testing examples tends to infinity, the SAC-BL algorithm performs well on abnormal data if the scores follow the generalized Gaussian-like distribution family.

First, the authors show that as the number of testing examples tends to infinity, the SAC-BL algorithm performs well on abnormal data if the scores follow the generalized Gaussian-like distribution family. But such conditions about the number of testing examples and the distribution of scores are overly restrictive for the real-world applications. So, the authors attempt to decrease the FPR of the SAC-BL algorithm under the condition of finite samples for practical anomaly detection. To this end, the authors redesign the BL algorithm by incorporating a randomization strategy and propose a novel stochastic BL (SBL) algorithm.

**Questions:**

Algorithmic content could be clearer, see:

https://shantoroy.com/latex/how-to-write-algorithm-in-latex/

For examples of algos, is that possible?

In the background, literature cited could focus more on limitations, is that possible?

Equations not of authors should be cited when needed?

I think the mathematics could be explained more in the text.

Experimental analysis is short, such as Table 1 being discussed,

More graphical results should be presented.

Future work could be discussed in conclusion or before

**Ethical Concerns:**

["NO or VERY MINOR ethics concerns only"]

**Final Justification:**

My ratings and decisions stand from the first round, while the author have addressed my comments I have still given the score that reflects the quality of the paper for NeurIPS.

**Limitations:**

More limitations could be expanded upon

**Paper Formatting Concerns:**

is the reference style correct? please check

**Quality:**

3

**Strengths And Weaknesses:**

Strengths:

Paper is well formatted, using LaTex formatting

References are Up to date, perhaps more could be added

Background materials are up to date and well presented



Weaknesses:

Algorithmic content could be clearer, see:

https://shantoroy.com/latex/how-to-write-algorithm-in-latex/

For examples of algos

In the background, literature cited could focus more on limitations

Equations not of authors should be cited

I think the mathematics could be explained more in the text.

Experimental analysis is short, such as Table 1 being discussed,

More graphical results should be presented.

Future work could be discussed in conclusion or before

---

> ### Author Rebuttal · Authors · 2025-07-30
>
> __Q1__: About the algorithm presentation.
>
> __Ans1__: Thank you for your suggestions regarding Algorithm 1. We will revise it based on the examples provided in the link you shared.
>
> __Q2__: In the background, literature cited could focus more on limitations, is that possible?
>
> __Ans2__: Thank you for your valuable feedback. We discuss the limitations of existing literature in the Introduction section. Prior work has primarily focused on designing or training score functions, while little attention has been paid to systematically investigating the false positive rate (FPR) of anomaly detection (AD) algorithms from a theoretical perspective. Notably, reducing FPR remains one of the most important yet challenging problems in the field (see [1] for details). Our work aims to fill this gap in the literature. In the new version, we will also include a discussion of these limitations in the Background section.
>
> [1] Pang, Guansong, et al. "Deep learning for anomaly detection: A review."
>
> __Q3,Q4__: About equations and mathematics in our submission.
>
> __Ans3__: Thanks for your suggestions about the equations and mathematics in our paper. We will  make revisions following your suggestions.
>
> __Q5, Q6__: About the experimental analysis and graphical results.
>
> __Ans4__: We appreciate your helpful suggestions regarding our experiments. In the new version, we will add detailed discussions and analyses of the results presented in Tables 1 and 2. Additionally, following your suggestions, we will provide graphical illustrations of the decision performance.
>
> __Q7__: Future work could be discussed in conclusion or before.
>
> __Ans5__:We sincerely appreciate your constructive feedback. Minimizing the FPR while controlling the FDR at a prescribed level is a valuable direction for future research. We will include relevant discussions in the new version.

---

> > ### Comment · Reviewer_aLKU · 2025-08-02
> > **Thanks for your comments**
> >
> > Thank you for adding your responses to my queries for the manuscript.

---

> ### Comment · Reviewer_aLKU · 2025-08-05
> **Thank you for your revisions**
>
> Thank you for reading my comments. I have read your reply and rebuttal comments for the comments I made and I am happy with the changes you have made, I have no more comments or suggestions at this time.

---

> > ### Author Response · Authors · 2025-08-07
> >
> > Thanks for your recognition of our work and rebuttal.

---

### Official Review · Reviewer_mvCm · 2025-06-26

**Clarity:** 3
**Significance:** 4
**Originality:** 3
**Rating:** 5
**Confidence:** 4

**Summary:**

This paper investigates the theoretical properties of the SAC-BL algorithm for visual anomaly detection, with a particular focus on its underexplored false positive rate (FPR). Specifically, the authors first demonstrate that the FPR of SAC-BL can be low under certain distributional assumptions. However, these assumptions may be too restrictive for real-world applications. To address this, the authors propose a new algorithm, SAC-SBL, which introduces stochasticity into the decision rule to reduce the FPR while maintaining the false discovery rate (FDR) control. Theoretical analysis shows that SAC-SBL achieves a lower FPR than SAC-BL almost surely. Experiments on MVTec, VisA, and BTAD datasets confirm consistent improvements in FPR and F1-score without compromising TPR. Overall, this paper is well-motivated, theoretically rigorous, and empirically validated, offering a novel perspective on the design of decision rules in anomaly detection.

**Questions:**

See weaknesses

**Ethical Concerns:**

["NO or VERY MINOR ethics concerns only"]

**Final Justification:**

All of my concerns have been well addressed. This paper offers a novel perspective on FPR and presents a method with strong theoretical guarantees and solid empirical performance. I will maintain my score for acceptance.

**Quality:**

3

**Strengths And Weaknesses:**

## Strengths
- Despite being theory-heavy, this paper is well-structured and readable. Motivations, algorithms, and results are presented clearly.
- This paper proposes a novel SAC-SBL algorithm, and provides solid theoretical results showing that the proposed method can decrease the FPR control while maintaining FDR, compared the SAC-BL algorithm.
- The proposed method is simple to implement and integrates naturally into existing SAC-BL frameworks.
- This paper validates the proposed method across multiple challenging datasets and at both the image and pixel levels, showing consistent performance gains in terms of reduced FPR and improved F1-score, with comparable or improved TPR.

## Weaknesses
I have not identify the major weaknesses, but I have some recommendations:
- It would be better if the title of the paper included the background of the research problem. For example, it could be revised to “On the SAC-BL Algorithm for Visual Anomaly Detection.”
- The authors should provide more detailed experimental setup.
- I am confused of the results of line 725 in Appendix, please give a more clear interpretations.

---

> ### Author Rebuttal · Authors · 2025-07-30
>
> __W1__: It would be better if the title of the paper included the background of the research problem. For example, it could be revised to “On the SAC-BL Algorithm for Visual Anomaly Detection.”
>
> __Ans1__: Thanks for your suggestions. We will rename the title of our submission as "On the SAC-BL Algorithm for Anomaly Detection".
>
> __W2__: The authors should provide more detailed experimental setup.
>
> __Ans2__: Our experimental settings follows [1].
> All images are resized to 256$\times$256 and normalized to the value in range of [0, 1]. The train batch size is 8. We use Pytorch 2.2 to implement our method and adopt AdamW to optimize it. Default training schedule is 700 epochs and initial learning rate is set to 0.0001. We conduct experiments on 8 NVIDIA GeForce GTX 4090 and 2 Intel(R) Xeon(R) Silver 4210 CPU @ 2.20GHz. Memory is 384GB. More details can be found in [1].
>
>
> [1] [2] Ma X, Wu J, Liu W. SAC-BL: A hypothesis testing framework for unsupervised visual anomaly detection and location[J].
>
> __W3__: I am confused of the results of line 725 in Appendix, please give a more clear interpretations.
>
> __Ans3__: Since $\Psi(\cdot ) \in \mathcal{F}$, we have  $\Psi(\rho_n) = \frac{1}{n^{\nu + o(1)}}$.
>
> Note that $\rho_n - \mu \to -\infty$, so $\Psi(\rho_n - \mu) \to 1$. It follows that $	(1-\eta)\Psi(\rho_n) + \eta  \Psi(\rho_n -\mu) \sim \frac{1}{n^{\beta}} $ where $\eta = \frac{1}{n^{\beta}}$. Similarly, the simple analysis can derive that
> $ 	O_{p}\left(\eta \upsilon_{n_1}\sqrt{\Psi(\rho_n-\mu)(1-\Psi(\rho_n-\mu))} \right) = o_{p}(\frac{1}{n^{\beta}})$ and  $O_{p}\left(  (1-\eta )\upsilon_{n_0}\sqrt{\Psi(\rho_n )(1-\Psi(\rho_n ))}\right)  = o_{p}(\frac{1}{n^{\beta}})$
> since $\kappa_{n_0} = o(\frac{1}{n^{\beta/2}}) $.

---

> > ### Comment · Reviewer_mvCm · 2025-08-03
> >
> > Thank you for your response. All of my concerns have been well addressed. This paper offers a novel perspective on FPR and presents a method with strong theoretical guarantees and solid empirical performance. Therefore, I believe the paper meets the standards of NeurIPS and recommend its acceptance.

---

### Official Review · Reviewer_tkwZ · 2025-06-29

**Clarity:** 3
**Significance:** 4
**Originality:** 3
**Rating:** 5
**Confidence:** 4

**Summary:**

This paper presents a theoretical analysis for the SAC-BL algorithm, focusing on its false positive rate (FPR), which has not been well-analyzed before. The authors propose SAC-SBL, a stochastic variant of SAC-BL, and demonstrate that it achieves a lower FPR while maintaining its false discovery rate (FDR) at under prescribed level.  Theoretical results are backed by experiments on various anomaly detection benchmarks.

**Questions:**

the calibrated set plays a key role in estimating empirical p-values. How does it influence the experimental results?

**Ethical Concerns:**

["NO or VERY MINOR ethics concerns only"]

**Quality:**

3

**Strengths And Weaknesses:**

Strengths
1, this paper provides a rigorous theoretical analysis for the asymptotic FPR of SAC-BL algorithm.
2, The proposed method is a well-motivated extension of SAC-BL, the motivation and contributions are clear.
3, The presentation is clear and the proofs in Appendix are sound.
4, Experimental results across three benchmarks show the improvements in terms of FPR and F1-score, adding practical credibility to the theory.

Weaknesses
1, The title lacks clarity about the application domain. Readers unfamiliar with SAC-BL may not recognize this as a paper on visual anomaly detection.
2, Apart from the Gaussian distribution, are there any other distributions that belong to the generalized Gaussian-like distribution family?
3, There exist some typos in line 198 of submission.
4, Do the results in Theorem 4.5 depend on the distributional assumptions in Definition 3.1?

---

> ### Author Rebuttal · Authors · 2025-07-30
>
> __W1__: The title lacks clarity about the application domain. Readers unfamiliar with SAC-BL may not recognize this as a paper on visual anomaly detection.
>
> __Ans1__: Thanks for your suggestions. We will rename the title of our submission as "On the SAC-BL Algorithm for Anomaly Detection".
>
> __W2__: Apart from the Gaussian distribution, are there any other distributions that belong to the generalized Gaussian-like distribution family?
>
> __Ans2__: Yes, the generalized Gaussian-like distribution family is quite
> general. We can easily verify that many popular and widely used distributions in theoretical analysis, such as exponential location distribution E(µ, 1), Laplace distribution L(µ, 1), Gaussian distribution N(µ, 1) and Gamma distribution G(2, 1), belong to the generalized Gaussian-like distribution family. We set the scale parameter to 1 in these distributions just
> for the analytical convenience.
>
> __W3__: There exist some typos in line 198 of submission.
>
> __Ans3__: Thanks for your careful review. We will correct the notation of $\mathcal{C}$  as "$\mathcal{C} = [ \frac{n} {1\alpha }, \frac{n} {2\alpha}, \cdots, \frac{n} { n  \alpha}  ]$".
>
> __W4__: Do the results in Theorem 4.5 depend on the distributional assumptions in Definition 3.1?
>
> __Ans4__: We clarify that Theorem 4.5 does not depend on the assumptions in Definition 3.1. All required conditions for the result are explicitly presented in Theorem 4.5.
>
> __Q1__: the calibrated set plays a key role in estimating empirical p-values. How does it influence the experimental results?
>
> __Ans5__: In practice, we observe that a larger calibrated set tends to yield better detection performance.

---

> ### Comment · Reviewer_tkwZ · 2025-08-03
>
> Thanks for the responses. My concerns are addressed.

---

### Official Review · Reviewer_gRFq · 2025-07-01

**Clarity:** 1
**Significance:** 2
**Originality:** 2
**Rating:** 4
**Confidence:** 3

**Summary:**

This paper improved the (Ma et al., 2025) by theoretically analyzing the FPR and proposing a stochastic version of the BL method. This paper discussed the decision rule of AD methods following (Ma et al., 2025), which is an important problem when applying AD to the real world. The author presented a theoretical guarantee and empirically showed the advantages over the original SAC-BL methods.

**Questions:**

- How to generalize the decision rule to any AD algorithm and any AD task? Since the p value is only calculated from the anomaly score, it should not be limited to the vision AD task and the reconstruction based SAC method.
- How to obtain the calibrated set? Is it a subset of trianing set or it requires the labeled anomaly?
- If the calibrated set is a subset, how to select the subset?
- how the hyper-parameters, such as alpha and the synthetic strategies affect the performance?

**Ethical Concerns:**

["NO or VERY MINOR ethics concerns only"]

**Final Justification:**

As other reviewers recognized, the FPR is more important.

The problem, threshold selection, itself is an important problem in AD applications. The author shows a clear respones to my main concern. So, I increase my score.

Although the expeirment is still not enough, the authors give a strong theoretical analysis.

**Limitations:**

The author does not discuss the potential negative societal impact.

**Paper Formatting Concerns:**

- There is too much content related to the background.
- The abstract is too verbose and it is simply a copy of sentences in the introduction.
- The section 4 needs carefully reorganization. The logic flow is weak through the whole section.

**Quality:**

2

**Strengths And Weaknesses:**

Strengths
- This work focuses on FPR, which is more practical.
- The proposed stochastic method is simple and effective.
- This paper discussed an important problem in UAD, but it is not ready for publication.

Weakness
- The content in total is an incremental work of (Ma et al., 2025) so far. Compared with (Ma et al., 2025), only an extended FPR analysis and a stochastic variant of BL are not enough for NIPS acceptance.
- It is not intuitive to see how the SBL can improve the FPR.
- It seems that the BL method relys on a calibrated set while how this set is constructed is unclear in the main text.
- No investigation on the related works. The authors should also compare the results with recent SOTA.
- see more comments in the question.

---

> ### Author Rebuttal · Authors · 2025-07-30
>
> __W1__: About our contributions.
>
> __Ans1__: We are sorry to confuse you about our  contributions. We would like to clarify our core contributions in more detail as follows:
> - A recent survey [1] highlights that existing anomaly detection (AD) methods—particularly unsupervised methods—often suffer from high false positive rates (FPR), and that reducing FPR remains one of the most important yet challenging problems in the field. However, to our best knowledge, there is no prior work that systematically investigates the FPR of AD algorithms from a theoretical perspective. Our work fills this gap in the literature.
> - Previous studies focus primarily on designing  scoring functions but neglect the decision algorithms. [2] is perhaps the first to explicitly identify this limitation and propose a new decision algorithm, the BL procedure, to address it. Since theoretical analysis of FPR requires a well-defined decision rule, our work builds upon the BL algorithm in [2].  Then, we introduce a statistical framework to analyze the performance of AD algorithm about FPR and establish rigorous  theories of BL procedure. Theoretical results show that BL procedure performs well as the number of testing samples tends to infinity.
> -  To further reduce the FPR under finite-sample conditions, we introduce a new stochastic strategy and propose the SBL algorithm. We also provide a theoretically justified simplified version of SBL to facilitate practical implementation. We prove that SBL achieves a lower FPR than BL while maintaining FDR control at a prescribed level. Finally, we validate the effectiveness of our method through extensive experiments across diverse anomaly detection benchmarks.
>
> Overall, [2] focuses mainly on a practical AD algorithm to improving AUROC, but our study aims to provide deeper theoretical insights into the FPR behavior of decision algorithms in AD. So, our research direction is fundamentally different from that of [2] and the contributions go well beyond an incremental extension of [2].
>
> [1] Pang, Guansong, et al. "Deep learning for anomaly detection: A review."
>
> [2] Ma X, Wu J, Liu W. SAC-BL: A hypothesis testing framework for unsupervised visual anomaly detection and location[J].
>
>
>
>
>
> __W2__: About our intuition.
>
> __Ans2__: Our core intuition is outlined in lines 194–202 of the submission, and we elaborate on it here for clarity. We emphasize that our method is theory-driven. Since there is a tradeoff
> between the detection performance of both normal and abnormal examples, we cannot only consider the TPR or FPR when designing the AD decision algorithm. Note that $FPR = \frac{FP}{FP+TN} = \frac{ |\mathcal{R}^c \cup \mathcal{H}_1|}{|\mathcal{H}_1|}$. where $\mathcal{R}$ is the set of indices of the rejected null hypothese, and  $\mathcal{R}^c$ is the complement of $\mathcal{R}$.So we can reduce the FPR by reducing FP, equivalent to increasing the number of rejecting null hypotheses $|\mathcal{R}|$.
> However, simply increasing rejections can inflate false discoveries, so we must control the FDR at a prescribed level. As shown in lines 121–130 of our submission, controlling FDR leads to a better trade-off between detection performance on normal and abnormal examples.
> So, our core intuition is to design a new stochastic strategy about p-values such that the decision algorithm can reject more null hypothesee than BL algorithm while satisfying the conditions for FDR control in [2]. The details about stochastic strategy can be found in line 203-216 of the submission.
>
> __W3, Q2, Q3__: About calibrated set.
>
> __Ans3__: We are sorry for the confusion regarding the  the calibrated set. In practice, we simply sample a small subset of normal examples from the training data to construct the calibrated set—no special procedure is required. The calibrated set is thus a subset of the training set and does not contain any anomaly data.  Since our work focuses on the unsupervised setting, no label information is needed for the calibrated set. In our experiments, we use 40% of the training data as the calibrated set.
>
> __W4__: About related work and and more comparison experiments with recent SOTA.
>
> __Ans4__: We would like to clarify that all significant relevant work has been cited and discussed in the Introduction and Background sections. To avoid any misunderstanding, we will include a dedicated Related Work section in new version.
> Following your suggestions, we have adopted several recent SOTA methods as baselines, including EfficientAD [3], RealNet [4], MambaAD [5], and DiAD [6]. The results, shown in Tables 1–3, consistently demonstrate the superior decision performance of our SAC-SBL method compared to these baselines.
>
> Table 1: the results on MVTech.
> |             |       | Image_level |          |       | Pixel_level |          |
> | ----------- | ----- | ----------- | -------- | ----- | ----------- | -------- |
> | Metric      | TPR   | FPR         | F1-Score | TPR   | FPR         | F1-Score |
> | EfficientAD | 97.52 | 18.51       | 79.86    | 93.85 | 18.33       | 86.51    |
> | RealNet     | 97.49 | 32.73       | 81.10    | 90.50 | 16.44       | 87.84    |
> | MambaAD     | 96.93 | 22.60       | 83.71    | 84.08 | 21.99       | 86.92    |
> | DiAD        | 95.90 | 41.32       | 80.76    | 92.38 | 18.37       | 88.74    |
> | SAC-SBL(ours)    | 98.91 | 11.79       | 88.19    | 95.93 | 9.03        | 97.75    |
>
>
> Table 2: the results on VisA.
> |             |       | Image_level |          |       | Pixel_level |          |
> | ----------- | :---: | ----------- | -------- | ----- | ----------- | :------: |
> | Metric      |  TPR  | FPR         | F1-Score | TPR   | FPR         | F1-Score |
> | EfficientAD | 95.97 | 64.17       | 73.48    | 94.13 | 44.19       |  85.04   |
> | RealNet     | 94.92 | 59.21       | 83.63    | 92.66 | 56.01       |  82.64   |
> | MambaAD     | 93.00 | 61.42       | 84.37    | 90.50 | 62.68       |  83.57   |
> | DiAD        | 95.75 | 56.44       | 89.01    | 92.05 | 45.02       |  89.63   |
> | SAC-SBL(ours)       | 98.95 | 51.92       | 94.95    | 95.12 | 33.83       |  98.88   |
>
> Table 3: the results on BTAD.
> |             |       | Image_level |          |       | Pixel_level |          |
> | ----------- | ----- | ----------- | -------- | ----- | ----------- | -------- |
> | Metric      | TPR   | FPR         | F1-Score | TPR   | FPR         | F1-Score |
> | EfficientAD | 94.38 | 41.04       | 62.71    | 96.39 | 83.41       | 88.61    |
> | RealNet     | 96.62 | 67.00       | 64.46    | 92.41 | 76.43       | 88.64    |
> | MambaAD     | 95.92 | 34.62       | 61.45    | 94.02 | 74.09       | 89.48    |
> | DiAD        | 96.71 | 40.94       | 67.20    | 89.11 | 80.30       | 81.36    |
> | SAC-SBL (ours)      | 99.67 | 30.65       | 73.33    | 97.54 | 66.98       | 97.61    |
>
>
> [3] Efficientad: Accurate visual anomaly detection at millisecond-level latencies.
>
> [4] Realnet: A feature selection network with realistic synthetic anomaly for anomaly detection.
>
> [5] Mambaad: Exploring state space models for multi-class unsupervised anomaly detection.
>
> [6] A diffusion-based framework for multi-class anomaly detection.
>
> __Q1__: About the generalization of SBL algorithm.
>
> __Ans5__: Note that our SBL algorithm imposes no restrictions on the architecture of  score function. Any score function can be directly integrated into SBL via empirical p-values, making it broadly applicable to anomaly detection (AD) tasks. Therefore, SBL is not limited to visual AD settings or reconstruction-based SAC methods. Additionally, the empirical p-values are computed based on scores of normal examples, rather than  anomaly scores.
>
> __Q4__: About the influence of $α$.
>
> __Ans6__: In statistics, $α$ is typically set to 0.05 and does not require adjustment in most cases. The experimental results on MVTech in Table 4 show that our SBL algorithm is not sensitive to $\alpha$.
>
> Table 4: the experimental results of sensitivity on MVTech.
> |       |       | image_level |          |       | pixel_level |          |
> | ----- | ----- | ----------- | :------: | ----- | ----------- | :------: |
> | alpha | TPR   | FPR         | F1-score | TPR   | FPR         | F1-score |
> | 0.01  | 98.92 | 11.77       |  88.13   | 95.94 | 9.03        |  97.74   |
> | 0.05  | 98.91 | 11.79       |  88.15   | 95.93 | 9.03        |  97.75   |
> | 0.1   | 98.91 | 11.79       |  88.16   | 95.93 | 9.03        |  97.75   |
> | 0.2   | 98.91 | 11.8        |  88.16   | 95.93 | 9.03        |  97.75   |
>
> Besides, The synthetic strategies are used solely for training the score functions. However, our paper aims to investigate the FPR of the decision algorithm. Thus,  the synthetic strategies are not the focus of our research. For details on these strategies, please refer to Section 4 of [2].

---

> ### Comment · Reviewer_gRFq · 2025-08-01
>
> Hi,
>
> Thanks for your responses.  While I understand the core contribution based on the responses and comments from other reviewers, I still have several minor concerns.
>
> - Is there any assumption on the calibrated set and the test set? It seems the calibrated set is assumed all normal, though it is very common in AD research. I would like to discuss the situation where those assumptions are not satisfied. For example, what if the calibrated set (as a subset of the training set) is kind of noisy? What if the test set contains too many anomalies that are very close to the normal sample. What is the worst case of the proposed threshold selection?
>
> - How the threshold of the baseline/SOTA methods is selected to calculate the evaluation metrics, since these methods do not use the BL algorithm?
>
> - Since the proposed algorithm can be applied to any AD methods, I suggest revising the inputs of Algorithm 1 to the anomaly score of train, calibrated, test, and synthetic. Then, the score of mix will become a concatenation of train and synthetic. **Hope to see the open-scource code shall it be accepted.**
>
> - Regarding the new title "On the SAC-BL Algorithm for Anomaly Detection", the experiments can be extended from 2 sides.
>     * adding more results applying SAC-BL to SOTA methods in a plug-and-play manner.
>     * adding more results for tabular or sequence data AD.
>     I am still looking forward to see these additional results in the final version. NOTE that this is only a suggestion and not expected to see results so far.
>
> - How to apply SAC-BL method to the growing datasets, i.e. the training set/testing set is frequently updated as new data is collected through time.

---

> > ### Author Response · Authors · 2025-08-01
> >
> > Dear gRFq,
> >
> > Thanks for your meaningful suggestions. Below, you will find answers and clarifications to your concerns.
> >
> > __Q1__: About the calibrated set in special cases.
> >
> > __Ans1__: Our method is applicable to any anomaly detection score function and makes no specific assumptions about the calibrated  set. In an unsupervised setting, only normal samples are available before the test phase. Consequently, the calibrated set is composed solely of normal samples.
> >
> >  Before discussing the situations in your comments, let us denote the score distribution of the original calibrated set as $\mathcal{D}_0$，and the distribution of anomaly samples in the test set as $\mathcal{D}_1$. We use $olp(\mathcal{D}_0, \mathcal{D}_1)$ to represent the length of the overlapping region between two distributions。We define the following two types of distributional perturbations:
> >
> > 1. Left-skewed Perturbation $P_L$：The perturbed distribution $P_L(\mathcal{D}_0)$ is shifted to the left relative to the original distribution, while the overlap $olp(P_L(\mathcal{D}_0),\mathcal{D}_1)$ remains unchanged.
> > 2. Right-skewed Perturbation $P_R$：The perturbed distribution$P_R(\mathcal{D}_0)$ is shifted to the right,resulting in an increased overlap $olp(P_R(\mathcal{D}_0),\mathcal{D}_1)$ .
> >
> > Now consider the following scenarios: When the calibrated set is contaminated with noise, both left-skewed and right-skewed perturbations may occur. In contrast, if the test set contains a large number of anomalies that are very similar to normal samples, right-skewed perturbation is more likely to arise.
> >
> > Our method is immune to left-skewed perturbation, as it does not affect the p-value. However, right-skewed perturbation increases the p-values of certain test samples, leading to more conservative decisions (i.e., a higher tendency to classify samples as normal), thereby impacting overall performance. The quality of the calibrated set affects the reliability of threshold selection in our method. A larger overlap between the calibration and anomaly distributions leads to degraded decision performance. It impacts decision performance by degrading the quality of the score distribution. This influence of right-skewed perturbation is 'uniform'—it also affects BL method. Despite this, our method consistently outperforms BL methods under such conditions, as demonstrated both theoretically and empirically in our paper.
> >
> >
> > __Q2__: How the threshold of the baseline/SOTA methods is selected to calculate the evaluation metrics?
> >
> > __Ans2__: For baseline methods, we adopt the empirical decision rule from previous literature, setting the threshold to achieve a TPR of 95% on the validation set.
> >
> > __Q3__: About the input of Algorithm 1 and more experiments.
> >
> > __Ans3__: Thanks for your valuable suggestions. We will make revisions following your suggestions.
> >
> > __Q4__: About the extension of SAC-BL method to the growing datasets.
> >
> > __Ans4__: The SAC-BL algorithm and our method  belong to the batch method. In our next work, we will develop a online or batch online extension of SAC-BL method, suitable for growing datasets.

---

> > > ### Comment · Reviewer_gRFq · 2025-08-03
> > >
> > > Thank you for your further answers. I have one final suggestion.
> > >
> > > I suggest authors to emphasis that FPR is important in real-world applications in the introduction section.

---

> ### Author Response · Authors · 2025-08-03
>
> We sincerely appreciate your valuable suggestion. We will emphasize the significance of the FPR in the Introduction section in the new version.

---

### Note · Authors · 2025-08-13

Dear SACs, ACs, and Reviewers,

Thank you for taking the time to consider our paper. We are pleased that our theoretical contributions have been recognized by the reviewers, and we have provided clarifications addressing the reviewers’ concerns during the rebuttal period.

## Important Remarks

During the rebuttal period, several key points from our exchanges with the reviewers are worth highlighting:
- All reviewers acknowledge our theoretical contribution in establishing the statistical analytical framework for the false positive rate (FPR) in the anomaly detection task.
- We clarify the details about the calibrated set and experimental settings.
- We provide additional experimental results using recent SOTA methods, which demonstrate the superiority of our method.


## Conclusion

In conjunction with the rebuttal, we summarize the core contributions of this paper as follows:

- We derive the mathematical relationship between FDR, TPR, and FPR, and highlight the advantages of FDR control.
-  We propose a statistical framework for analyzing the FPR performance of anomaly detection algorithms and provide rigorous theoretical foundations for the BL procedure.
-  We present a novel stochastic strategy and propose the SBL algorithm, which can theoretically achieve a lower FPR than the BL procedure while maintaining FDR control at a specified level. Comprehensive experimental results further confirm the effectiveness of the proposed method.

Given these factors, we believe our work would make a valuable contribution to NeurIPS. Thank you for your time and consideration!

---

### Decision · Program_Chairs · 2025-09-17

**Decision:**

Accept (poster)

**Comment:**

This paper provides a theoretical analysis of the strong anomaly constraint (SAC) network and a betting-like (BL) algorithm, focusing on the false positive rate (FPR) in anomaly detection. The reviewers agree that the paper makes a meaningful theoretical contribution. Building on this analysis, the authors propose a simple yet effective extension of SAC-BL by introducing stochasticity. The theoretical analysis shows the advantage of the proposed method. Experimental results further demonstrate its effectiveness. The author response successfully addressed the reviewers’ concerns, and the additional experiments included during the rebuttal phase further strengthen the paper.